# Human Synovia Contains Trefoil Factor Family (TFF) Peptides 1–3 Although Synovial Membrane Only Produces TFF3: Implications in Osteoarthritis and Rheumatoid Arthritis

**DOI:** 10.3390/ijms20236105

**Published:** 2019-12-03

**Authors:** Judith Popp, Martin Schicht, Fabian Garreis, Patricia Klinger, Kolja Gelse, Stefan Sesselmann, Michael Tsokos, Saskia Etzold, Dankwart Stiller, Horst Claassen, Friedrich Paulsen

**Affiliations:** 1Friedrich Alexander University Erlangen-Nürnberg (FAU), Institute of Functional and Clinical Anatomy, 91054 Erlangen, Germany; judithpopp@gmx.de (J.P.); martin.schicht@fau.de (M.S.); fabian.garreis@fau.de (F.G.); Patricia.Klinger@gmx.de (P.K.);; 2University Hospital Erlangen, Department of Trauma Surgery, 91054 Erlangen, Germany; kolja.gelse@web.de; 3University of Applied Sciences Amberg-Weiden, Institute for Medical Engineering, 92637 Weiden, Germany; stefansesselmann@web.de; 4Charité-Universitätsmedizin Berlin, Institute of Legal Medicine and Forensic Sciences, 10117 Berlin, Germany; michael.tsokos@charite.de (M.T.); saskia.etzold@charite.de (S.E.); 5Martin Luther University Halle-Wittenberg (MLU), Department of Legal Medicine, 06108 Halle (Saale), Germany; dankwart.stiller@uk-halle.de; 6Martin Luther University Halle-Wittenberg (MLU), Department of Anatomy and Cell Biology, 06108 Halle (Saale), Germany; horst.claassen@fau.de; 7Sechenov University, Department of Topographic Anatomy and Operative Surgery, 119146 Moscow, Russia

**Keywords:** trefoil factor family peptides (TFF), synovial membrane (SM), synovial fluid (SF), osteoarthritis (OA), rheumatoid arthritis (RA)

## Abstract

Objective: Trefoil factor family peptide 3 (TFF3) has been shown to support catabolic functions in cases of osteoarthritis (OA). As in joint physiology and diseases such as OA, the synovial membrane (SM) of the joint capsule also plays a central role. We analyze the ability of SM to produce TFF compare healthy SM and its secretion product synovial fluid (SF) with SM and SF from patients suffering from OA or rheumatoid arthritis (RA). Methods: Real-time PCR and ELISA were used to measure the expression of TFFs in healthy SM and SM from patients suffering from OA or RA. For tissue localization, we investigated TFF1-3 in differently aged human SM of healthy donors by means of immunohistochemistry, real-time PCR and Western blot. Results: Only TFF3 but not TFF1 and -2 was expressed in SM from healthy donors as well as cases of OA or RA on protein and mRNA level. In contrast, all three TFFs were detected in all samples of SF on the protein level. No significant changes were observed for TFF1 at all. TFF2 was significantly upregulated in RA samples in comparison to OA samples. TFF3 protein was significantly downregulated in OA samples in comparison to healthy samples and cases of RA significantly upregulated compared to OA. In contrast, in SM TFF3 protein was not significantly regulated. Conclusion: The data demonstrate the production of TFF3 in SM. Unexpectedly, SF contains all three known TFF peptides. As neither articular cartilage nor SM produce TFF1 and TFF2, we speculate that these originate with high probability from blood serum.

## 1. Introduction

In vertebrates, joints have evolved into highly conserved organs that are essential for locomotion. Their joint capsules are lined with a synovial membrane (SM), which secretes synovial fluid (SF) and is therefore essential for maintaining the functionality of the joint connections. SM is the innermost membrane of the joint capsule in diarthroses, consisting of an intimal layer comprising 1–2 cells and a subintimal layer containing particularly blood vessels, adipocytes, fibroblasts, and macrophages [1]. The intimal layer comprises two different cell types: type A-cells (type-A-synoviocytes), which belong to macrophage lineage, repopulated from local precursors [2] and their main function is phagocytosis of debris from the joint cavity [3]; and type B-cells (type-B-synoviocytes) which are fibroblast-like cells that produce and secrete hyaluronan [4], lubricin (PRG4) [5], and surface-active phospholipids [6] among others, as essential components of SF. The SF primarily functions as a biological lubricant, provides nutrients as well as regulatory cytokines, and reduces friction and wear of the participating joint structures during movement [7]. 

The mammalian trefoil factor (TFF) family comprises three protease-resistant peptides of 7–12 kDa—TFF1, TFF2, and TFF3 (also ITF)—that have a distinct highly conserved motif of 6 cysteine residues in common, the so-called trefoil domain [8]. TFF peptides are mainly expressed in mucous epithelia (mucosa) performing cell type-specific interactions with mucins [9]. Several studies have described TFFs primarily as secretory peptides of the gastrointestinal tract [10]. However, these were later also found in various other tissues all over the body [11]. 

The functions of the three known TFF peptides are as diverse as the tissues producing them [12]: TFF peptides are important for the maintenance and repair of intestinal mucosa. They influence the mucous viscosity [13] as well as the surface mucosal defense and protect the mucous barrier [14]. They induce cell migration and are, therefore, motogens in restitution, an important mechanism of epithelial repair, also in bone repair [15]. Furthermore, TFF peptides act anti-apoptotic, are involved in immune response [16], promote angiogenesis and have healing effects (for review see [17,18]). 

In a previous study, we found that TFF peptides are absent in healthy articular cartilage, but in certain disease states, such as osteoarthritis (OA) or septic arthritis, the TFF family member TFF3 supports catabolic functions and links inflammation to tissue remodeling processes, which distinguishes TFF3 as a potential factor in the pathogenesis of OA [19]. 

OA is a chronic degenerative joint disease characterized by cartilage degradation and is the most common joint disease around the world, causing an increasing burden on health systems [20]. Inflammatory symptoms, especially synovitis, usually occur as secondary manifestations during the progression of the disease. Rheumatoid arthritis (RA) is, in contrast to OA, a primary inflammatory joint disease characterized by chronic inflammation, destruction of articular cartilage, and subarticular bone accompanied by the loss of joint function. About 0.5–1% of the global population are affected, and it is a major cause of disability [21]. SM and SF are involved in both OA and RA. An overview of the role of synovitis in OA, for example, the production of pro-inflammatory cytokines [22] or synovial hypertrophy and hyperplasia [23], has been summarized by Sellam and Berenbaum [24]. Concerning RA, SM is the main tissue affected by the disease [25], and synovial fibroblasts play a pivotal role in the pathogenesis [26]. 

With regard to the above-mentioned findings, we question in our present study whether TFF peptides are also expressed in SM and secreted into SF. Another question is whether TFFs play a role in the pathogenesis of OA and RA by being up- or downregulated under disease conditions in comparison to healthy tissues and SF.

## 2. Results

### 2.1. TFF3, but not TFF1 and TFF2, is Expressed in Human SM of Healthy Donors and SM of Patients Suffering From OA or RA on the Protein and RNA Level

#### 2.1.1. Presence and Localization of TFFs in the Synovial Membrane

Gene expression of TFF1, TFF2, and TFF3 of healthy, osteoarthritis (OA), and rheumatoid arthritis (RA) affected SM was analyzed by semi-quantitative PCR (Figure 1A). Specific amplification products were only detected for TFF3 (≈302 bp) in all sample groups (healthy, OA, and RA), whereas TFF1 and -2 were absent in each healthy, OA, and RA SM. mRNA of the human stomach served as a positive control for all three TFFs and β-actin as the loading control.

Immunohistochemistry was performed on 5 µm sections of healthy human SM from differently aged donors (Figure 1B). Using specific antibodies to TFF1, TFF2, and TFF3, reactivity in SM (indicated by a red reaction product) could be visualized only for TFF3 and was especially strong in type B synoviocytes of SM. In addition, intense TFF3 immunoreactivity was also present in the walls of blood vessels within the subsynovial layer. There was no obvious difference in the slides of the three differently aged donors. No antibody reactivity was observed for TFF1 or TFF2 in any of the samples.

#### 2.1.2. Quantification of TFF3 in the Synovial Membrane

TFF3 gene expression was analyzed in human SM from healthy donors and in samples of patients suffering from OA and RA by means of quantitative PCR. Our data revealed inter-individual differences within each sample group. Although the results showed a slight upregulation in the disease-affected samples comparing OA or RA (Figure 2A) with SM from healthy donors, this difference was not statistically significant.

ELISA was only performed for TFF3 in SM as this was the only TFF peptide detectable in healthy or disease-affected SM. For this, the protein amount of TFF3 in healthy human SM was measured and compared to SM of patients suffering from OA or RA revealing a protein amount of 298.4 pg/mg in the healthy group (ranging from 46.5 to 663.0 pg/mg), 473.7 pg/mg (ranging from 141.1 to 1125.3 pg/mg) in case of OA, and 299.3 pg/mg (ranging from 155.9 to 701.8 pg/mg) in case of RA. Compared to the healthy samples, the protein amount of TFF3 in RA was nearly the same, whereas in OA, it was 1.6-fold higher (not statistically significant) (Figure 2B and Table 1). 

### 2.2. All Three TFF Peptides (TFF1, -2 and -3) Are Detectable in Healthy Synovial Fluid As Well as in SF of Patients Suffering From OA or RA

#### Western Blot

Presence of TFF1, TFF2, and TFF3 was analyzed by Western blot in SF from healthy donors as well as in SF of patients suffering from OA and RA. We used commercial and duplicated antibodies against TFFs. Results revealed antibody reactivity in all samples including human stomach, which served as a positive control for all three TFFs. Distinct protein bands for all samples were detected for TFF1 at around 55 kDa and in healthy SF samples at 25 kDa. For TFF2 and TFF3, we detected protein bands at around 55, 37, 30, and 23 kDa (Figure 3A). Furthermore, TFF3 was detectable at about 14 kDa in a healthy as well as in an RA sample matching the dimeric form of TFF3. For TFF1, we performed the Western blot under reducing conditions with DTT and β-ME as reducing agents. We did not detect the monomeric forms of any of the three TFFs in SF samples, which would have resulted in protein bands at about 7, 35, and 40 kDa (TFF1) or 12 kDa (TFF2) and 7 kDa for TFF3. The positive control stomach confirmed the result for all TFFs. 

### 2.3. TFF2 Protein Concentration Is Significantly Increased in Synovial Fluid of Patients Suffering From RA, TFF3 Protein Concentration Is Reduced in Cases of OA and RA, and TFF1 Shows No Change

#### ELISA

ELISA was performed in order to measure the protein concentration of TFFs in healthy human SF and SF of patients suffering from OA or RA (Figure 3B–D). 

TFF1 was detected in all samples except for one case of healthy SF and was not significantly changed between healthy (50.9 pg/mg, ranging from 0.8 to 258.1 pg/mg) and disease-affected conditions (OA: 31.2 pg/mg, ranging from 9.5 to 117.3 pg/mg; and RA: 33.0 pg/mg, ranging from 18.8 to 67.7 pg/mg) (Figure 3B and Table 2). TFF2, which was also present in all samples, revealed a low protein concentration in healthy samples (13.7 pg/mg, ranging from 2.4 to 74.3 pg/mg) that was hardly changed in cases of OA (9.0 pg/mg, ranging from 0.4 to 45.9 pg/mg), but was strongly increased in cases of RA (203.5 pg/mg, ranging from 0.3 to 835.4 pg/mg) (Figure 3C and Table 3). TFF2 protein concentration was 22.6-fold higher in RA in comparison to OA (statistically significant) and a 14.9-fold higher concentration in RA compared to healthy samples (not statistically significant). TFF3 was significantly downregulated (2.6-fold) in OA samples of SF (2833.9 pg/mg, ranging from 1730.8 to 4828.8 pg/mg) in comparison to healthy ones (7807.0 pg/mg, ranging from 2214.6 to 29279.2 pg/mg). With regard to the relation of OA and RA, the relative protein concentration of TFF3, there was a significantly lower (1.4-fold) upregulation in RA samples (4083.9 pg/mg, ranging from 2051.8 to 8413.0 pg/mg) than in OA samples (2833.9 pg/mg, ranging from 1730.8 to 4828.8 pg/mg) (Figure 3D and Table 4). A comparison between all three TFF peptides revealed that the protein concentration of TFF3 in total was much higher than the concentration of TFF1 (127.8-fold) or TFF2 (65.1-fold).

## 3. Discussion

In the past, we described TFF3 production by human articular chondrocytes if the cartilage was affected by osteoarthritis (OA) with chondrocytes of healthy articular cartilage having been TFF3-negative [19]. Furthermore, our findings show that the pro-inflammatory cytokines tumor necrosis factor α (TNFα) and interleukin-1β (IL-1β) induce TFF3 gene expression in cultured primary articular chondrocytes. In addition, there is upregulation of distinct matrix metalloproteinases (MMP), which are well known as cartilage degrading enzymes in primary cultured chondrocytes after stimulation with recombinant human (rh) TFF3 [19]. Stimulation with rhTFF3 revealed a proapoptotic effect by increasing caspase 3/7 activity. These observations have led to the conclusion that TFF3 supports catabolic functions in human articular cartilage and is a possible factor in the pathogenesis of OA. Bijelic et al. investigated a possible role of TFF3 during endochondral ossification in mice. They demonstrated that TFF3 is not present in areas of resting healthy growth plate cartilage but can be visualized in areas of endochondral ossification. The localization of TFF3 was similar to that reported in OA cartilage, suggesting similar roles for TFF3 during the two different processes [27].

OA is not only restricted to articular cartilage but affects the entire joint structure, including especially the synovial membrane (SM) which takes part in the etiology and the progression of the disease [28]. Our herein presented results reveal the presence of TFF3 on mRNA and protein level in SM of OA and RA affect knee joints and, in contrast to articular cartilage, also in healthy SM. This finding was confirmed by also detecting TFF3 on the protein level and localizing studies in healthy human SM of differently-aged healthy donors. 

Unexpectedly, our data revealed the expression of not only TFF3 but also TFF1 and TFF2 in synovial fluid (SF) of healthy individuals and patients suffering from OA and RA, leading to the question of the source of TFF1 and TFF2. It seems obvious that the detected TFF3 is expressed and secreted into SF by SM. This is in accordance with the semi-quantitative PCR and immunohistochemistry results revealing the presence of TFF3 in SM and suggesting an active secretion of TFF3 into SF. In addition, articular chondrocytes might also be able to contribute to the relatively high TFF3 content in SF as they are able to produce TFF3 [19] whereas TFF1 and TFF2 are not produced in articular cartilage chondrocytes, neither in healthy articular joint structures nor in OA or RA affected joint structures. One possible explanation for the presence of TFF1 and -2 in SF could be the fact that SF does not exclusively consist of proteins produced by SM or other joint structures. As an ultrafiltrate of blood plasma, it is primarily composed of proteins (including mucins) and small molecules such as glucose, urea, and electrolytes that are derived from plasma in the blood vessels of the articular joint capsule [29,30] and transported through the SM acting as a kind of selective filter [7]. This mechanism of diffusion from blood plasma through SM into the joint cavity is also possible for TFF peptides, and it would explain the small amount of TFF1 and -2 in comparison to TFF3 in SF as the latter is additionally secreted into SF by SM and probably also by articular chondrocytes [31]. Moreover, it has been demonstrated for other body fluids, such as, e.g., saliva [32] or blood serum [31], that the concentration of TFF3 is regularly higher than that of TFF1 and -2. 

Nevertheless, it cannot be ruled out that the detection of TFF1 and -2 in SF samples might also be based on unspecific binding or cross-reactive antibody reactions. The effect of heterophilic antibody reactions on the reliability of assays has been investigated in several studies [33]. Hampel et al. evaluated and discussed their effect in assays for chemokine and cytokine levels in OA and RA synovial fluid [34] and, in particular, detected false-positive reactions for chemokines in the assay of RA synovial fluid samples. Furthermore, Samson et al. validated commercial assays for measurements of TFFs in serum [35]. They revealed acceptable results for TFF2- and TFF3-commercial assays but not for TFF1 assays which showed a poor precision and a narrow measuring range. An absolute reliable identification of the protein bands detected in Western blot analysis is only possible by means of protein sequencing and will be a matter of further investigation.

Taking a closer look at the molecular structure of TFF peptides, there is, besides the six cysteine residues that form intramolecular disulfide bonds determining the TFF domain, an additional seventh cysteine at the C-terminal end of the amino acid chain that allows TFF1 and TFF3 to form intermolecular homo- or heterodimers [17,36,37]. We detected several distinct protein bands at different molecular weights. This is contrary to our expectations as we performed Western blot under reducing conditions. The detected difference in TFFs protein size implies additional protein modifications. We, therefore, expected only distinct bands at about 7 (TFF1 and -3) and 12 kDa (TFF2) representing TFF monomers. The findings suggest a possible presence of non-reducible bounds between the TFF peptides and their binding partners or posttranslational modifications of the TFF peptides in SF. Several studies exist that indicate specific protein bands for TFF1, -2, and -3 heterodimers at higher molecular weights [38]. SF of OA and RA comprises a huge amount of different proteins that could be able to interact with TFF peptides performing, for example, disulfide bonds [34]. It is unclear so far which proteins are able to directly interact with the TFF peptides in SF leading to heterodimers or complexes. This will be the subject of future studies as well as determining a possible function of these interactions in healthy synovial joint structures and OA and RA.

Hypothetically, one option could be the modification of the viscosity of SF. This effect has already been demonstrated for TFF peptides by Thim et al. in mucin solutions [13]. It is well known that TFF peptides and mucins are specifically co-expressed in mucin-producing epithelia, especially of the gastrointestinal tract. Here, for example, TFF3 interacts with MUC5AC in the human stomach and with MUC2 in the duodenum while TFF2 is co-produced with MUC6 in the human stomach and duodenum [39]. TFF1 is also bound to MUC5AC in human gastric mucosa [40]. The exact molecular mechanisms of these interactions, however, are still not understood. Currently, there are only a few studies concerning the expression of mucins in synovial tissue. Violin et al. detected an upregulated mucin expression, particularly MUC3, and also the expression of MUC5AC in synovial tissues of OA and RA affected human synovial joints in comparison to healthy tissues [41]. In contrast to Violin et al., a newer study also found MUC1 to be present in synovial membrane cells as well as mononuclear cells in RA synovial tissues, but not in OA, hypothesizing that mucins may also play a role in immunoinflammatory reactions in the pathogenesis of RA [42]. Nevertheless, further investigations are required to make a clear statement about the possible functions of mucins in OA and RA synovial fluid.

Our data reveal a downregulation of TFF3 between healthy and OA (statistically significant) as well as RA SF samples and an upregulation of TFF3 between healthy and OA (not statistically significant) SM samples. RA SM samples showed no significant changes. With regard to articular cartilage chondrocytes, the increased production of TFF3 in SM is in accordance with the already mentioned results of Rösler et al. concerning the increased TFF3 expression in articular cartilage under inflammatory conditions (OA and septic arthritis). Inflammation of SM usually happens during the progress of OA [22], but there are also studies that indicate the existence of SM inflammation also in earlier stages of OA as a possible factor in the pathogenesis. Benito et al. detected, for example, a higher expression of TNFα and IL-1β as well as a higher cell infiltration and vascularization in the synovial tissue of patients with early OA compared to advanced OA [43]. This leads to the hypothesis that synovial membrane inflammation might occur before the first signs of cartilage degradation. The samples we used for real-time PCR and ELISA were obtained from patients who underwent total knee arthroplasty. This is usually performed in late stages of OA or RA meaning a high level of cartilage degradation already present but not necessarily a high level of synovial membrane inflammation. This could be one explanation for the differences of TFF3 seen in SM and SF. However, all this is speculative, and so far, the different measurements are purely descriptive as, in other tissues, chronic inflammation or trauma have been shown to be associated with a downregulation of TFF3. Thus, Chaiyarit et al. detected a significant reduction of TFF3 expression in oral mucosa of patients with chronic periodontitis compared to healthy mucosa [44], and Siber-Hoogeboom et al. described significantly lower TFF3 saliva protein concentrations in cases of rhonchopathy and obstructive sleep syndrome (OSA) compared to a healthy control group [45]. 

Although both diseases, OA and RA, show signs of synovial membrane inflammation, most studies that compared synovial membrane inflammation in OA and RA found higher expression levels of cytokines and a higher number of infiltrating immune cells in RA tissues [46]. Looking at the results of Rösler et al., who detected an induction of TFF3 gene expression due to stimulation of cultivated primary chondrocytes with TNFα and IL-1β [19], and considering the fact that TNFα and IL-1β are well known as important mediators of inflammation [47,48] it seems possible that this mechanism is the reason for the significantly higher level of TFF3 in RA synovial fluid. 

It is clear to us that this study is descriptive and that the number of samples used is rather small. However, it should be emphasized that this is a first description of TFF in the synovial membrane and synovia and that it is extremely difficult to obtain healthy synovial membranes and especially synovia from "healthy" people. Beside the observed differences of TFF3, we detected a highly significant upregulation of the TFF2 protein concentration in SF from patients suffering from RA whereas nearly no regulation was found in the case of OA. As mentioned above, the composition of TFF peptides in SF might also be influenced by the blood serum levels of the TFF peptides. Therefore, it would be important to have knowledge about the human donors of SF and to know whether these suffered from any additional diseases that could have had an influence on the serum levels. One example for this is diabetes mellitus type 1. Barrera Roa et al. demonstrated that TFF3 expression is regulated by insulin and glucose [49]. Serum TFF3 levels are downregulated in serum of diabetes mellitus type 1 patients, and the presence of glucose and insulin results in elevated serum levels of TFF3. Similar effects have been reported by Vestergaard et al. who detected elevated serum levels of TFF1 and -3 in patients suffering from inflammatory bowel disease [50]; and a study of Viby et al. found elevated serum levels of TFF1, -2 and -3 in patients with COPD [51]. Furthermore, a higher serum level of TFF3 was detected in patients with chronic kidney disease, metastatic and secondary carcinoma, and acute gastroenteritis [52]. In addition, TFF2 serum concentrations were recently shown to increase in chronic kidney disease [53]. Moreover, treatment of the patients with anti-inflammatory medication might also have an effect on the expression of TFF peptides. Koitabashi et al. detected an increased TFF2 expression in the gastric cancer cell line MKN45 after incubation with indometacin, a non-steroidal anti-inflammatory drug [54]. As our RA patients have received previously medication (we have not evaluated this), we cannot exclude contributory effects. All these mechanisms could influence the levels of TFF1, -2, and -3 in SF and limit the findings of the present study. Nevertheless, our findings may contribute to a better preparation and collection of sample data (medication, additional diseases), for example, when analyzing other ultrafiltrates of blood plasma like urine, saliva, or lacrimal fluid.

## 4. Materials and Methods

### 4.1. Human Tissues/Fluids

All performed investigations followed the Declaration of Helsinki for research involving human tissue and with approval by the local ethics committee. Informed consent was obtained from each patient before surgery.

#### 4.1.1. Synovial Membrane

Healthy synovial membrane (SM) (*n* = 14) was obtained, with institutional review board approval, by the Department of Legal Medicine, Charité Berlin, Germany of male and female donors aged at least 3 years and no more than 92 years who were autopsied in cases of sudden and violent death. The samples were used for reverse transcriptase-polymerase chain reaction (semi-quantitative PCR), real-time PCR, immunohistochemistry, and ELISA.

SM of patients suffering from osteoarthritis (OA) (*n* = 10) or rheumatoid arthritis (RA) (*n* = 10) were obtained from male and female patients aged between 56–81 (OA) and 38–84 (RA) years, respectively, who underwent total knee arthroplasty at the Department of Trauma Surgery, University Hospital Erlangen, Germany. These samples were used for conventional semi-quantitative PCR, real-time PCR, and ELISA.

After removal, SM samples were frozen in liquid nitrogen and after addition of triton buffer (300 µl) and protease and phosphatase inhibitors (10 µl/1 ml each) the samples were homogenized in a Speedmill Plus (Analytik Jena AG, Jena, Germany) followed by incubation for 30 min on ice. Before use, the samples were centrifuged at 13000 rpm for 20 min at 4 °C. The samples were then incubated on ice for 30 min. The supernatants were frozen and then analyzed by enzyme-linked immunosorbent assay (ELISA).

#### 4.1.2. Synovial Fluid

Healthy human synovial fluid (SF) (*n* = 13) was obtained from donors, 4 female and 9 male, aged between 29–80 years by the Department of Legal Medicine, Halle, Germany from autopsy cases of an unnatural manner of death.

OA and RA SF (each *n* = 20) came from two different sources: ten OA samples (7 female and 3 male, aged 64–87 years) and ten RA samples (9 female and 1 male, aged 39–76 years) of SF were obtained from patients who were undergoing total knee arthroplasty at the Department of Trauma surgery, University Hospital Erlangen, Germany. Another ten OA samples (6 female and 4 male, aged 65–84 years) and ten RA samples (6 female and 4 male, aged 57–77 years) were provided by the Department of Molecular Immunology, Friedrich Alexander University Erlangen-Nürnberg, Germany. The diagnosis of OA was based on clinical and radiographic evaluations according to standard criteria, and the diagnosis of RA was made from patients fulfilling the American College of Rheumatology/European League Against Rheumatism classification criteria [55]. Each patient gave informed consent prior to surgery, and the institutional ethics committee approved the study (Ref.No. 3555; FAU Erlangen Nürnberg). The SF samples were transferred directly into a 1.5 ml Eppendorf tube and immediately frozen on dry ice. All samples were then transported on dry ice from the clinic to the laboratory and stored at –80 °C until analysis. The samples were centrifuged for 5 minutes at 13,000 rpm before use. The supernatants were frozen or subsequently analyzed by Western blot and enzyme-linked immunosorbent assay (ELISA). 

#### 4.1.3. Human Control Tissue

A tissue sample of the human stomach was obtained from a body donor (male, 83 years) donated to the Institute of Anatomy and Cell Biology, Martin Luther University Halle-Wittenberg, Germany who died by natural cause and did not suffer from any affection of the gastrointestinal tract.

### 4.2. RNA Isolation and Complementary DNA (cDNA) Synthesis

Frozen samples of SM were homogenized with Speedmill Plus (Analytik Jena AG, Jena, Germany) and total RNA from healthy, OA, and RA samples of human SM was purified with isopropanol and repeated ethanol precipitation and isolated with Superscript™ II-RT-Kit (Thermo Fisher Scientific Inc., Waltham, MA, USA) using the following protocol: Contaminating DNA was eliminated by digestion of RNAse free DNAse I (30 min at 37 °C). DNAse was inactivated after adding 50 mM EDTA and incubating 10 min at 65 °C. The RNA concentration was measured spectroscopically (ND 1000; NanoDrop Technologies, Wilmington, DE, USA). This procedure was followed by performing cDNA synthesis using RevertAid H Minus reverse transcriptase (Thermo Fisher Scientific Inc., Waltham, MA, USA) and following a standard protocol. β-actin was used as an internal control to ensure the integrity of cDNA.

### 4.3. Reverse Transcriptase-Polymerase Chain Reaction (Semi-Quantitative PCR)

cDNA of healthy, OA, and RA samples was amplified performing semi-quantitative PCR with specific primer pairs for TFF1, -2, and -3 (see Figure 1) as previously described by Siber-Hoogeboom [45]. To estimate the amount of amplified PCR product, a β-actin PCR was performed as the loading control. Positive control (human stomach) and negative control (replacement of cDNA by RNA-free water) were included in the investigation.

### 4.4. Quantitative Real-Time (qPCR)

The expression level of TFF3 in healthy (in total *n* = 12, each 2 samples were pooled to *n* = 6), OA (*n* = 5) and RA (*n* = 5) SM was analyzed after RNA isolation (as described above) by quantitative real-time PCR (qPCR) usingthe Quant-Studio 12 K Flex Real-Time PCR System (Life Technologies, Carlsbad, CA, USA) with TaqMan probes (5′ - Fam - CTGTCTGCAAACCAGTGTGCCGT - Tamra - 3′) as fluorescent dye in the gene expression assay Verso 1-step RT-qPCR ROX Mix (# AB-4101/A; Thermo Fisher Scientific Inc., Waltham, MA, USA). Verso 1-step RT-qPCR ROX Mix is an assay that performs whole qPCR (including cDNA-synthesis) in a single step assay. For detailed information and cycling protocol, see the product sheet. The relative amount of target mRNA of TFF3 was calculated and normalized to that of HPRT (hypoxanthine-guanine-phosphoribosyltransferase) mRNA (gene expression assay HS03929096_g1; Life Technologies, Carlsbad, CA, USA) functioning as endogenous control, as HPRT has, similar to TFF3, a low expression level. For quantitation, the relative standard curve method was performed, and the relative gene expression of TFF3 in healthy samples was afterward compared to OA and RA sample groups. The used primers are listed in Table 5. 

### 4.5. Western Blot Analysis

TFFs were analyzed by Western blot in healthy, OA, and RA synovial fluid. The samples were diluted with PBS (phosphate-buffered saline), and protein concentration was spectroscopically measured by Bradford protein assay. Each sample (30 µg protein) was mixed with reducing sample buffer (RSB) containing 0.5 M dithiothreitol (DTT), 50% glycerin, 0.05% bromphenolblau, 20% β-mercaptoethanol (β-ME) and boiled for 5 minutes. SDS-PAGE and Western blot analyses were performed as described by [56]. The membrane was incubated with primary antibodies to TFF1 (dilution 1:100), TFF2 (dilution 1:70), and TFF3 (dilution 1:400) at 4°C overnight before applying the secondary antibodies (dilution 1:4000) conjugated to horseradish peroxidase for 2 h and detecting the bands by chemiluminescence using ECL Plus (Amersham Pharmacia, Uppsala, Sweden). Human stomach served as a positive control for all three peptides. The molecular weights of the detected protein bands were estimated referring to a standard protein marker (PageRuler Prestained Protein Ladder #26616; Thermo Fisher Scientific Inc., Waltham, MA, USA) ranging from 10 to 170 kDa. The used antibodies are listed in Table 6. Alpha-1-antitrypsin and actin were used as internal control (primary antibody dilution 1:2000, secondary antibody dilution 1:4000).

### 4.6. Immunohistochemistry

Samples of human SM (fixed in 4% formaldehyde) were paraffin-embedded, and immunohistochemistry was performed on 5 µm sections of the tissue with antibodies to TFF’s. For antigen retrieval, the sections were cooked in citrate buffer (pH = 6.0) for 5 minutes. To inhibit nonspecific binding, the sections were incubated with normal serum for 20 minutes using normal goat serum for TFF1 and TFF3 and normal rabbit serum for TFF2 (Dako, Glostrup, Denmark, dilution: 1:20 with TBST). Sections were incubated with the primary antibodies (dilution: 1:100 (TFF1, -2), 1:300 (TFF3) with TBST) in a humidified box overnight at 4 °C and with the biotinylated secondary antibodies (dilution 1:200 with TBST) for 1 h. For visualization of the antibody reaction, AEC + high sensitivity substrate chromogen (K3469, Dako, Glostrup, Denmark) was used for about 2–3 minutes depending on the used antibody. The results were examined using light microscopy (Biorevo microscope BZ-9000, Keyence, Osaka, Japan). Both positive (tissue slides of the stomach) and negative (incubating slides only with secondary antibody) controls were included in immunohistochemistry. The used primary and secondary antibodies are listed in Table 6.

### 4.7. Enzyme-Linked Immunosorbent Assay (ELISA)

For quantitative detection of TFFs in healthy (*n* = 13), OA (*n* = 20), and RA (*n* = 20) human synovial fluid and of TFF3 in healthy (*n* = 10), OA (*n* = 10), and RA (*n* = 10) human synovial membrane the following ELISA-kits from USCN Life Science Inc., Wuhan, PRC were used: SEB049Hu (TFF1), SEA748Hu (TFF2), and SEB656Hu (TFF3). Their detection range was 3.12–200 pg/ml (TFF1), 31.2–2000 pg/ml (TFF2) and 125–8000 pg/ml (TFF3). The color reaction was analyzed with a microplate spectrophotometer (Kinetic Microplate Reader Vmax, Molecular Devices, Sunnyvale, CA, USA) by quantifying the absorbance spectrophotometrically at a wavelength of 405 nm (TFF1) or 450 nm (TFF2, -3). The concentration of TFF1, -2, and -3 (pg/mg) in each sample was then determined by comparing their optical density to the standard curve and the prior measured amount of total protein in each sample.

### 4.8. Statistical Analysis of ELISA and Real-Time PCR Data

The Kolmogorov–Smirnov test was used for testing the normal distribution of the ELISA and real-time PCR data. This was followed by performing the Mann–Whitney *U*-Test (not normally distributed data) or ANOVA with Bonferroni correction (normally distributed data) and the results were interpreted using InStat statistical software (GraphPad Software, San Diego, CA, USA). Values of *p* < 0.05 were considered statistically significant.

## 5. Conclusions

In summary, our results demonstrate the presence of trefoil factor family peptide 3 (TFF3) in synovial membrane and unexpectedly show that synovial fluid contains all three TFF peptides that originate with high probability from blood serum. Significant changes in the TFF peptide concentration of synovial fluid between healthy individuals and patients suffering from OA and RA suggest a possible role in the pathogenesis of both diseases. However, to get further insights into possible functions of TFF peptides in healthy individuals and cases of OA or RA, we need future studies with more background information about the patients under investigation.

## Figures and Tables

**Figure 1 ijms-20-06105-f001:**
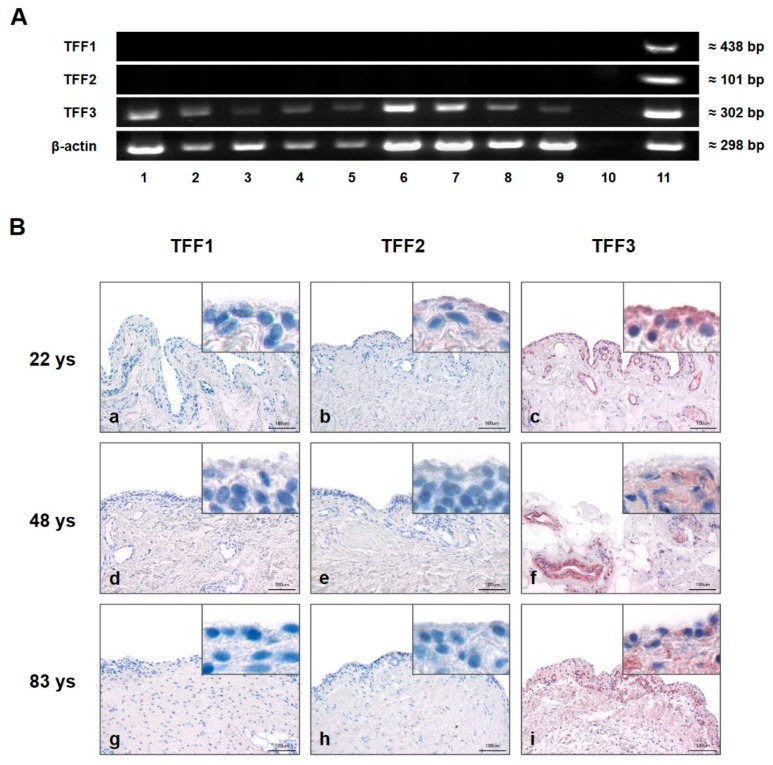
Presence and localization of TFFs in synovial membrane (SM). (**A**) Semi-quantitative PCR analysis of TFFs mRNA in human SM of five healthy (28, 48, 54, 78, and 92 years (Lines 1, 2, 3, 4, and 5)), rheumatoid arthritis (RA) (Lines 6 and 7) and osteoarthritis (OA) (Lines 8 and 9) samples. Line 10 represents the negative control (without cDNA template), and human stomach serves as positive control (Line 11). Beta-actin (β-actin) was used as the loading control. (**B**) Immunohistochemical analysis of TFF1, -2 and -3 in human SM of 22 (**a**,**b**,**c**), 48 (**d**,**e**,**f**), and 83 (**g**,**h**,**i**) year-old healthy donors. TFF3 is present in each examined sample (**c**,**f**,**i**). TFF1 (**a**,**d**,**g**) and TFF2 (**b**,**e**,**h**) reveal negative results irrespective of the donor’s age. Insets show magnifications. Scale bars: 100 μm. Red staining indicates positive antibody reaction.

**Figure 2 ijms-20-06105-f002:**
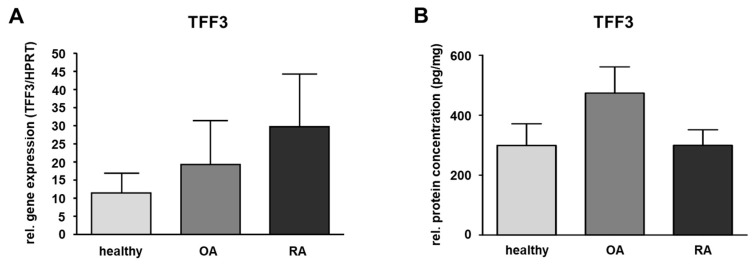
Quantification of TFF3 in the synovial membrane (SM). (**A**) SM samples of healthy (*n* = 6) and OA (*n* = 5) as well as RA (*n* = 5) affected knee joints. The relative gene expression is expressed in TFF3/HPRT and visualized as mean value and standard error of the mean (SEM). Mean values are 11.38 (healthy), 19.29 (OA,) and 29.61 (RA). No significant difference is detected in relative gene expression between healthy and OA or healthy and RA samples performing the Mann–Whitney U-test (significance level *p* ≤ 0.05). (**B**) TFF3 protein level in human SM of healthy (*n* = 10), OA (*n* = 10) and RA (*n* = 10) samples were detected by ELISA. Mean values are: 298.4 pg/mg (healthy), 473.7 pg/mg (OA), 299.3 pg/mg (RA). The protein concentration is expressed in pg/mg and visualized as mean value and standard error of the mean (SEM). The protein amount of TFF3 in the three groups showed no statistically significant difference performing the ANOVA with Bonferroni correction (significance level *p* ≤ 0.05).

**Figure 3 ijms-20-06105-f003:**
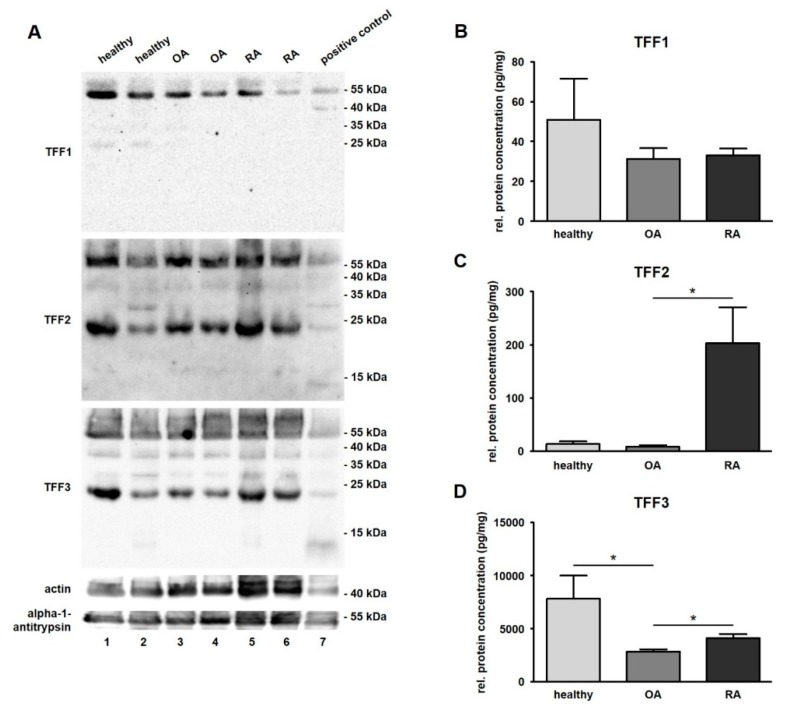
Detection and quantification of TFF peptides in synovial fluid (SF). (**A**) Western blot analysis of TFFs in human SF of healthy (Line 1 and 2), osteoarthritis (OA; Line 3 and 4), and rheumatoid arthritis (RA; Line 5 and 6) samples. Proteins of the human stomach serving as positive control (Line 7) were included in the test. Actin (≈ 43 kDa) and alpha-1-antitrypsin (≈ 51 kDa) were used as loading control. Molecular weight marker is shown on the right. ELISAs of TFF1 (**B**), –2 (**C**), and –3 (**D**) in human SF of healthy (*n* = 13), OA (*n* = 20) and RA (*n* = 20) samples. Mean values are: TFF1: 50.87 pg/mg (healthy), 31.16 pg/mg (OA), and 33.04 pg/mg (RA); TFF2: 13.69 pg/mg (healthy), 8.97 pg/mg (OA), 203.50 pg/mg (RA); TFF3: 7807.29 pg/mg (healthy), 2833.90 pg/mg (OA), 4083.92 pg/mg (RA). The protein concentration is expressed in pg/mg and visualized as mean value and standard error of the mean (SEM). * indicates significant differences performing the Mann–Whitney *U*-Test (significance level *p* ≤ 0.05).

**Table 1 ijms-20-06105-t001:** ELISA quantification of TFF3 in the synovial membrane.

Relative Protein Concentration of TFF3 in Human Synovial Membrane (SM)		
Samples	Mean (pg/mg)	Range (pg/mg)
healthy (*n* = 10)	298.4	46.5–663.0
OA (*n* = 10)	473.7	141.1–1125.3
RA (*n* = 10)	299.3	155.9–701.8

**Table 2 ijms-20-06105-t002:** ELISA quantification of TFF1 in synovial fluid.

Relative Protein Concentration of TFF1 in Human Synovial Membrane (SM)		
Samples	Mean (pg/mg)	Range (pg/mg)
healthy (*n* = 13)	50.9	0.8–258.1
OA (*n* = 20)	31.2	9.5–117.3
RA (*n* = 20)	33.0	18.8–67.7

**Table 3 ijms-20-06105-t003:** ELISA quantification of TFF2 in synovial fluid.

Relative Protein Concentration of TFF2 in Human Synovial Membrane (SM)		
Samples	Mean (pg/mg)	Range (pg/mg)
healthy (*n* = 13)	13.7	2.4–74.3
OA (*n* = 20)	9.0	0.4–45.9
RA (*n* = 20)	203.5	0.3–835.4

**Table 4 ijms-20-06105-t004:** ELISA quantification of TFF3 in synovial fluid.

Relative Protein Concentration of TFF3 in Human Synovial Membrane (SM)		
Samples	Mean (pg/mg)	Range (pg/mg)
healthy (*n* = 13)	7807.0	2214.6–29279.2
OA (*n* = 20)	2833.9	1730.8–4828.8
RA (*n* = 20)	4083.9	2051.8–8413.0

**Table 5 ijms-20-06105-t005:** Primers used for semi-quantitative PCR and real-time PCR.

Primers Used for Semi Quantitative PCR				
Template	Sense 5´-3´	Antisense 5′-3′	Product	Annealing Temperature
**TFF1**	TTTGGAGCAGAGAGGAGG	TTGAGTAGTCAAAGTCAGAGCAG	438 bp	60 °C
**TFF2**	GTGTTTTGACAATGGATGCTG	CCTCCATGACGCACTGATC	110 bp	59 °C
**TFF3**	GTGCAAGCCAAGGACAG	CGTTAAGACATCAGGCTCCAG	302 bp	56 °C
**β-actin**	GATCCTCACCGAGCGCGGCTACA	GCGGATGTCCACGTCACACTTCA	298 bp	60 °C
**Primers used for real-time PCR**				
**TFF3**	TCCAGCTCTGCTGAGGAGTA	CAGTCCACCCTGTCCTTG	self-provided	62 °C

**Table 6 ijms-20-06105-t006:** Primary and secondary antibodies used for western blot (WB) and immunohistochemistry (IHC)

Antibodies	Method	Specifity	Source
**rabbit anti-TFF1**	IHC	monoclonal	AJ1765a, Abgent
**rabbit anti-TFF1**	WB	polyclonal	orb100742, Biorbyt
**goat anti-TFF2**	IHC, WB	polyclonal	SP (P-19): sc-23558, Santa Cruz
**mouse anti-TFF2**	WB	monoclonal	H00007032-M01, Abnova
**rabbit anti-TFF3**	IHC	polyclonal	ABIN 669786, Bioss Inc.
**rabbit anti-h-TFF3**	WB	polyclonal	provided by Prof. W. Hoffmann, Magdeburg
**rabbit anti-actin**	WB	polyclonal	Actin (H-300): sc-10731, Santa Cruz
**rabbit anti-alpha-1-antitrypsin**	WB	polyclonal	A 0012, Dako
**goat anti-rabbit**	IHC	polyclonal	BA-1000, Vector Laboratories Inc.
**rabbit anti-goat**	IHC	polyclonal	E 0466, Dako
**goat anti-rabbit**	WB	polyclonal	P 0448, Dako
**goat anti-mouse**	WB	polyclonal	IgG-HRP: sc-2005, Santa Cruz
**donkey anti-goat**	WB	polyclonal	IgG-HRP: sc-2020, Santa Cruz

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
