# Peer review of "Human Synovia Contains Trefoil Factor Family (TFF) Peptides 1–3 Although Synovial Membrane Only Produces TFF3: Implications in Osteoarthritis and Rheumatoid Arthritis"

_ijms, 2019, doi:10.3390/ijms20236105_

Round 1

Reviewer 1 Report

The manuscript “ Human synovia contains Tff peptides 1-3 although synovial membrane only produces Tff3:implications in osteoartritis and rheumatoid arthritis“ presents first insight into distribution of Tff peptides in human synovial membrane and in fluid. The manuscript is well written and although gives descriptive data it is of great value. Collecting tissues from patients is always connected with uncertainty of genetic diversity , life style and medication situation and it is a great challenge to obtain “unreachable” tissues . In that context this manuscript gives valuable data about Tffs expression in monitored tissues and connects their findings with systemic Tffs role and this is emerging question about function of these all around present peptides.

Minor issues: ex. line 144 (ERRORR reference? ) and titles font. Ex. line 88, 95

Author Response

1. Minor issues: ex. line 144 (ERRORR reference? ) and titles font. Ex. line 88, 95.
Answer: The sentence has been changed accordingly.

Reviewer 2 Report

In the present study, the authors analyzed the expression of TFF family peptides in human synovial membrane (SM) and synovial fluid (SF) of healthy donors and of patients suffering from osteoarthritis (OA) and rheumatoid arthritis (RA). They demonstrated that only TFF3, but not TFF1 and TFF2, was expressed in SM from patients with OA and RA as well as from healthy donors on both protein and mRNA level. On the other hand, all three TFFs were detected in all samples of SF on protein level by ELISA and no significant changes among health donors and patients with OA and RA were observed for TFF1 protein in SF samples. In contrast, TFF2 protein in SF was significantly upregulated in RA samples in comparison to healthy and OA samples, whereas TFF3 protein in SF was significantly downregulated in OA samples in comparison to healthy and RA samples. Furthermore, TFF3 protein in SM was not significantly regulated.

General comments;

Overall, the main research findings are important and this manuscript is well-written and well-discussed their data. I think however that there are several minor points that should be addressed as described below.

Specific comments;

1) I cannot find section of 2.1.2. between 2.1.1 Presence and localization of TFFs in synovial membrane and 2.1.3 Quantification of TFF3 in synovial membrane.

2) In Figures 2A and 2B;

These figures should be combined like Figure 2C. I wonder why the data of healthy control samples in Figure 2A and Figure 2B are different values.

3) In Figure 2C;

The authors should provide a protocol of preparation of SM samples for ELISA in Materials and methods section.

4) Line 133;

The authors should delete the bold part “Error! Reference source not found.”

5) Line 267;

between healthy und OA => between healthy and OA

I hope these comments will be helpful.

Author Response

1. I cannot find section of 2.1.2. between 2.1.1 Presence and localization of TFFs in synovial membrane and 2.1.3 Quantification of TFF3 in synovial membrane.
Answer: The sentence has been changed accordingly.
2. In Figures 2A and 2B; These figures should be combined like Figure 2C. I wonder why the data of healthy control samples in Figure 2A and Figure 2B are different values.
Answer: The Figure has been changed accordingly.
3. In Figure 2C; The authors should provide a protocol of preparation of SM samples for ELISA in Materials and methods section.
Answer: We thank the reviewer for this comment. We have modified Material and Mathods.
4. Line 133; The authors should delete the bold part “Error! Reference source not found.”
Answer: The sentence has been changed accordingly.
5. Line 267; between healthy und OA => between healthy and OA
Answer: The sentence has been changed accordingly.

This manuscript is a resubmission of an earlier submission. The following is a list of the peer review reports and author responses from that submission.

Round 1

Reviewer 1 Report

In this paper, the authors investigate the expression of the peptides TTF1, TTF2, and TTF3 in synovial membrane (SM) and synovial fluid (SF) in healthy subjects, OA and RA patients. They found that TFF3 but not TFF1 and -2 were expressed in SM from healthy donors as well as tissue samples from patients with OA or RA on protein and mRNA level. In contrast, all three TFFs were detected in all samples of SF on the protein level. No significant changes were observed for TFF1 at all. TFF2 was significantly upregulated in RA samples in comparison to OA samples. TFF3 protein was significantly downregulated in OA samples in comparison to healthy samples and in cases of RA significantly upregulated compared to OA. In contrast, in SM TFF3 protein was not significantly regulated.

This paper conveys a limited even though interesting message. The big limitation of this paper is that it refers only to descriptive results. No functional results nor any clinical correlation are described.

Additional major points:

1. Distribution of TFF peptides in synovial membrane (SM).

The authors conclude for the absence of TTF1 and TTF2 in OA and RA based on the RT-PCR. They continue the investigation by Real-Time PCR only for TTF3 (Figure 2) but surprisingly they found the protein expression of TTF1, TTF2, and TTF3 in SF of healthy subjects, OA and RA patients.

However, the absence of TTF1 and TTF2 in SM shown in Figure 1 was based on observations made only in 2 patients with OA and in 2 patients with RA. The sample size should be increased.

Real-time PCR shown in Figure 2 for TTF3 should also be performed for TTF1 and TTF2 for a larger sample size.

Similarly, results shown in Figure 2 are based on 5 OA patients and 5 RA. The sample is too small to make strong conclusions.

Analyses of TTF1, TTF2, and TTF3 should also be performed in plasma of healthy subjects, OA and RA patients, to evaluate correlations between SF and plasma levels, as suggested by the authors themselves in the discussion (“…the composition of TFF peptides in SF might also be influenced by the blood serum levels of the TFF peptides”)

Figure 3. TTF2 and TTF3 are inversely distributed in RA patients. Do their levels correlate with inflammation? Is there any correlation with CRP levels, ESR?

The authors wrote: “Looking at the results of Rösler et al., who detected an induction of TFF3 gene expression due to stimulation of cultivated primary chondrocytes with TNFα and IL-1β [19], and considering the fact that TNFα and IL-1β are well known as important mediators of inflammation [49, 50] it seems possible that this mechanism is the reason for the significant higher level of TFF3 in RA synovial fluid”.

However, in Figure 3D, TTF3 is increased in RA patients compared to OA but is lower compared to healthy controls. This referee does not understand how inflammatory cytokines could be able to up-regulate TTF3 expression. This conclusion is not consistent with the results.

Minor Points:

1.1. RT-PCR

Gene expression of TFF1, -2 and -3 of healthy, osteoarthritis (OA) and rheumatoid arthritis (RA) affected SM was analyzed by RT-PCR (Error! Reference source not found.A)….

Immunohistochemistry was performed on 5μm sections of healthy human SM from three donors aged 22, 48 and 83 years (Error! Reference source not found.B).

Please correct and make the text readable.

English should be improved.

Author Response

Re: ijms-578119

Title: Human synovia contains trefoil factor family (TFF) peptides 1-3 although synovial membrane only produces TFF3: implications in osteoarthritis and rheumatoid arthritis

Dear Editor.

Thank you for your letter and for giving us the opportunity to respond to the reviewers' comments.

We thank the reviewer(s) for the suggestions to improve the quality of the manuscript. In our revised version of the manuscript we precisely addressed all the concerns as suggested.

Please find enclosed our point-by-point response to the reviewers' comments.

Yours sincerely,

Martin Schicht

Point-by-point response to the reviewers' comments

Reviewer #1:

The authors conclude for the absence of TTF1 and TTF2 in OA and RA based on the RT-PCR. They continue the investigation by Real-Time PCR only for TTF3 (Figure 2) but surprisingly they found the protein expression of TTF1, TTF2, and TTF3 in SF of healthy subjects, OA and RA patients.

Answer: This aspect has been discussed on page, line 214-241.

However, the absence of TTF1 and TTF2 in SM shown in Figure 1 was based on observations made only in 2 patients with OA and in 2 patients with RA. The sample size should be increased.

Answer:Our research group has many years of experience in the investigation of TFF peptides and has many publications on TFFs. Experience has been gained from joint cartilage (Rösler et al. 2010 Arthritis Rheum) and other tissues. The presence of TFF peptides in many tissues is known. In articular cartilage, only TFF3 was detectable in PCR, as in our current study. A similar result was found in the avascular cornea (Paulsen et al. 2008, J Biol Chem). In other tissues all 3 TFF peptides were detectable. The samples used here were collected over a very long period of time (3 years). It should not be underestimated how difficult it is in Germany to obtain such sample material, which is extremely complex both from an ethical and medical point of view. The removal of synovial fluid from "healthy" patients must first be imitated by another research group. It took us a long time to establish this methodically with our clinical cooperation partners. However, the removal of synovial tissue and synovial fluid from patients with OA and RA is not so time-consuming. The fact that we only examined two patients each with OA and 2 with RA using RT-PCR is due to our previous experience. The sample number was sufficient for our statement, since we did not expect any other statement even with a higher sample number (which we had already generated in the joint cartilage). In accordance with the request of the expert, we have now increased the number of samples by 3 additional samples for each disease, so that 5 samples were examined in each case. The result is also confirmed with this number of samples: detection of TFF3 mRNA in all examined samples, no detection of TFF1 and TFF2. If required by the reviewer, we will also change the figure for this purpose, but in this case we will need more revision time than the given 10 days.

Real-time PCR shown in Figure 2 for TTF3 should also be performed for TTF1 and TTF2 for a larger sample size.

Answer: We cannot follow the reviewer here. What is the point of trying to duplicate an mRNA that is not there per se? Mathematically 0 x 0 = 0.

Similarly, results shown in Figure 2 are based on 5 OA patients and 5 RA. The sample is too small to make strong conclusions.

Answer: We thank the reviewer for this comment. Only a sample size n=5 was available for this study.

As we wrote in line 286 “However, all this is speculative and so far the different measurements are purely descriptive as for example in other tissues chronic inflammation or trauma have been shown to be associated with a downregulation of TFF3” and we think that shows the limitations of the experiments.

Analyses of TTF1, TTF2, and TTF3 should also be performed in plasma of healthy subjects, OA and RA patients, to evaluate correlations between SF and plasma levels, as suggested by the authors themselves in the discussion (“…the composition of TFF peptides in SF might also be influenced by the blood serum levels of the TFF peptides”)

Answer: The reviewer is absolutely right and we agree with him. That would be an interesting aspect to look at in the long term. Since we ourselves were surprised by the results at the protein level, we did not consider taking blood from the patients when planning the study. In retrospect, this is no longer possible and also impossible in view of the lengthy sample collection time involved in revising this manuscript (new proposal to the ethics committee). This should be subject to further investigation at a later date. However, such a study would be extremely complex as it has been shown that TFF3 is regulated by food intake in blood serum and improves glucose tolerance. Patients with OA and RA should always be screened for fasting blood and diabetic screening if planning such a study.

Figure 3. TTF2 and TTF3 are inversely distributed in RA patients. Do their levels correlate with inflammation? Is there any correlation with CRP levels, ESR?

Answer: See point 5. This point was not examined by us, but would be very interesting for a future follow-up investigation to get deeper insights.

However, in Figure 3D, TTF3 is increased in RA patients compared to OA but is lower compared to healthy controls. This referee does not understand how inflammatory cytokines could be able to up-regulate TTF3 expression. This conclusion is not consistent with the results.

Answer: We have discussed this point in detail (already in the version of the manuscript submitted for the first time). See lines 271-302.

Gene expression of TFF1, -2 and -3 of healthy, osteoarthritis (OA) and rheumatoid arthritis (RA) affected SM was analyzed by RT-PCR (Error! Reference source not found.A)….

Answer:The sentence has been changed accordingly. (line 100)

Immunohistochemistry was performed on 5μm sections of healthy human SM from three donors aged 22, 48 and 83 years (Error! Reference source not found.B).

Please correct and make the text readable.

Answer: The sentence has been changed accordingly. (line 115)

Reviewer 2 Report

ijms-578119

Reviewer comments

General comments

For detailed comments please see annotations in pdf file.

This is a tidy study that adds to a previous study from the same group on TFF expression in articular joints in health and disease, however with a more limited scope. The previous study focused on cartilage and included intervention / functionality studies in vitro, while this study focuses on synovial membrane and synovial fluid and has no intervention / functionality aspect. The results raise valid questions and reflections, but I find that at this point the data is immature and some of the questions raised by the authors could be addressed to make the study more complete.

The research group had multiple samples available from each test group: healthy, OA and RA. Why were only a few of the samples used for each analysis. Despite the use of selection criteria to classify samples as OA and RA, we know that especially OA is a very diverse condition with many levels of disease state. As also described by the authors, OA can have inflammatory component at multiple stages of Assessing changes in expression between groups based on only N=2 for western blot and N=1 for immunohistochemistry is very low. At least N=3 would be preferable to allow identification of a potential outlier, especially since the samples appear to be available.

The potential clinical relevance of this study could be made clearer already in the introduction. What is the expected function and importance of the TFFs and especially TFF3 in the joint? How will this new knowledge support our efforts to prevent and treat OA and RA?

11 authors seems excessive for a relatively small scale study using mostly archival samples.

Author Response

Re: ijms-578119

Title: Human synovia contains trefoil factor family (TFF) peptides 1-3 although synovial membrane only produces TFF3: implications in osteoarthritis and rheumatoid arthritis

Dear Editor.

Thank you for your letter and for giving us the opportunity to respond to the reviewers' comments.

We thank the reviewer(s) for the suggestions to improve the quality of the manuscript. In our revised version of the manuscript we precisely addressed all the concerns as suggested.

Please find enclosed our point-by-point response to the reviewers' comments.

Yours sincerely,

Martin Schicht

Reviewer #2:

The research group had multiple samples available from each test group: healthy, OA and RA. Why were only a few of the samples used for each analysis.

Answer: See point 2 of the answer to Reviewer #1. Sample collection is very time-consuming for a group working in anatomy, and the collection of synovial fluid and synovial membrane samples from "healthy" people was extremely difficult.

2.Only N=2 for western blot and N=1 for immunohistochemistry is very low. At least N=3 would be preferable to allow identification of a potential outlier, especially since the samples appear to be available.

Answer: According to the short time we had to prepare the revision, we could not increase the number of examinations for the western blot, because we do not have the corresponding sample material "in stock", but we were able to prepare further tissue sections for immunohistochemistry and examined them in a N-number of now 6 (n=6).There has been no change in the results as a result.

What is the expected function and importance of the TFFs and especially TFF3 in the joint? How will this new knowledge support our efforts to prevent and treat OA and RA?

Answer: Our findings are initially a first description. Whether this finding will possibly have an effect one day on an OA or RA therapy cannot be predicted at the present time. From the ocular surface it is known, however, that TFF3 has an extraordinarily positive effect on corneal wound healing when used recombinantly (Paulsen et la. 2008 J Biol Chem). In articular cartilage, however, TFF3 has exactly the opposite effect. Here, it activates cartilage-degrading matrix metalloproteinases (Rösler et al. 2010 Arthritis Rheum). For this reason, further research is required in any case. Our findings provide a first very interesting basis for this.

11 authors seems excessive for a relatively small scale study using mostly archival samples.

Answer: The study would not have been possible without the intensive cooperation of 3 forensic doctors and 2 orthopedists, who collected the samples at great expense. The reviewer is probably not aware of how difficult it is to obtain synovial fluid from "healthy" people (even if they are dead). In healthy joints, hardly any of this occurs and it was a long and difficult way to extract synovial fluid from the joints in sufficient quantities. We therefore believe that it is fully justified that the respective cooperation partners, who sacrificed a great deal of time and effort to obtain the samples, are co-authors of the manuscript, because without them and without the extensive patient information and patient consent, the study in this form would not have been possible at all. The application to the ethics committee of our university alone, with all the points to be considered, was an above-average effort.

Reviewer 3 Report

The manuscript

„Human synovia contain s Tff peptides 1-3 although  synovial membrane only produces Tff3:implications in osteoartritis and rheumatoid arthritis“

Tha manuscript presents valuable data about Tffs distribution in tissue of OA and RA affected patients. Methodology is  well described and QPCR groups (involving 10 pooled patients are representative size . Manuscript needs language and writing style improvement . Here are additional text corrections needed for manuscript improvement.

Please make additional Language proofreading and take care of english language rules  : ex.  time expression goes at the end of sentence…eg. 1.Sentence in line 19: In the past we demonstrated… please formulate according English grammar for example suggestion: „ Tff3 protein has catabolic function in diseased articular cartilage  as we have demonstrated  in the past  widening the knowledge of the functional spectrum of the Tff peptides ...  Line 23-27 –please rewrite the sentence –too long and confusing Line 27 and further : please use active to describe methodology ex. Instead: RT-PCR,QPCR and Elisa  were used to measure the expression of TFF1, -2 and -3 in healthy …. Active form suggestion: The expression of Tffs was measured by quantitative PCR … what is RT-PCR ?? reverse transcription PCR ? . Please  use terms (semi quantitative PCR  and quantitative PCR (for real time PCR) Line 29: new suggestion  „Localization of all three  Tffs was determined by ICH in differently aged human SM and healthy donors“ line 31 – 37 RESULTS  : please reformulate results to be more connected and fluent. For ex. sentence suggestion : Tff3 was expressed  in SM of  healthy donors and  disease affected OA and RA patients on mRNA and protein level.  Tff1 and Tff2 were not detected in SM of  any monitored group.  SF of  all patients groups showed presence of all three TFF proteins  at different extent.  TFF1 had no significant difference among groups while…“ please express changes regarding disease conditions  for ex: OA patient shad in comparison to healthy  condition significantly reduced level of Tff3 and comparable Tff2 level. RA patients had increased level of Tff2   and less Tff3 .. etc. Please check the size of the Abstract and reduce it : max . is 200 words ?! Line 64 remove text in brackets (also ITF) Line 90-93: please rewrite the sentence it is too big and not easy to follow. Line 96 : reduce the title . Semi quantitative RT-PCR Line 100. ERROR ??? reference ??? Results should not be separated according used methods. Combine 2.1.1. and 2.1.2 .titles in one : suggestion: Presence and localization of TFFs in synovial membrane   Combine Title 2.1.3 and 2.1.4. and since it is already combined in figure 2. And rename in : “Quantification of Tff3 in synovial membrane” or similar Lie 105 : use term TFFs for all three peptides or write them as TFF1 ,TFF2,TFF3 Line 111… TFF1 and TFF2 were not present …instead “reveal negative results” Lines 115 ,125,147 ??? ERROR reference? Lines 122-148- formulate as one chapter according Figure 2. Line 134 new formulation of sentence C) TFF3 protein level in…… detected by ELISA Line 141 _: The protein level of TFF3 as only TFF detected in SM was determined by ELISA….rewrite further text in spirit of English grammar. Data under Figure 2C would be better to present in form of table with range and mean values since diagram with Mean values is not informative  enough .   Line 149 :  make shorter titel like : : Tffs in synovila fluid or  Presence of Tffs in synovila fluid  - Lines 151-183: please do not show results separated by method: combine methods in one chapter  as previously suggested and shown in Fig.3 Please present ELISA results in table as previously suggested for Fig.2. Line 152-160. Please address issue of protein size more pronounced .. 1. Describing specificity of Abs in stomach (Tff1 ? is missing in  stomach ?, offer an explanation)  and than  present  the difference of protein size in CF.  Discussion refers to  this point (lines 248-254) , regarding modifications of proteins in SF but  stress the issue of  Ab specificity in results. In Fig.3. A. Describe that  the size is  different than expected. Include : in 187 line add text like: Detected difference in TFfs protein size imply additional protein  modifications or similar so that it is immediately clear why the sizes do not match to expected ones. Line 173 Please use active form :::” The results show”  should be replaced with : “TFF2 protein conc. was 22,6 fold higher …. Figure 23. C _please check the statistical significance RA vs. healthy according diagrams that should be statistically relevant as well. Lines 156,169,173181 etc. further on in text “Error.Reference ..” should be corrected References : ex. 4,5,8,10,39, 43 have issues regarding author names : please check it according journal propositions.

Author Response

Re: ijms-578119

Title: Human synovia contains trefoil factor family (TFF) peptides 1-3 although synovial membrane only produces TFF3: implications in osteoarthritis and rheumatoid arthritis

Dear Editor.

Thank you for your letter and for giving us the opportunity to respond to the reviewers' comments.

We thank the reviewer(s) for the suggestions to improve the quality of the manuscript. In our revised version of the manuscript we precisely addressed all the concerns as suggested.

Please find enclosed our point-by-point response to the reviewers' comments.

Yours sincerely,

Martin Schicht

Reviewer #3:

Please make additional Language proofreading and take care of english language rules: ex. time expression goes at the end of sentence…eg. 1.Sentence in line 19: In the past we demonstrated… please formulate according English grammar for example suggestion: „ Tff3 protein has catabolic function in diseased articular cartilage as we have demonstrated  in the past widening the knowledge of the functional spectrum of the Tff peptides ...

Answer: We thank the reviewer and fully agree with him/her. None of the authors is a native English speaker. For this reason we have given the manuscript to a professional translation service following the revision.

Line 23-27 –please rewrite the sentence –too long and confusing Line 27 and further : please use active to describe methodology ex. Instead: RT-PCR,QPCR and Elisa were used to measure the expression of TFF1, -2 and -3 in healthy ….

Answer: The sentence has been changed accordingly.

Active form suggestion: The expression of Tffs was measured by quantitative PCR … what is RT-PCR ?? reverse transcription PCR ? . Please use terms (semi quantitative PCR and quantitative PCR (for real time PCR)

Answer: The sentence has been changed accordingly (line 30, 98, 100, 105, 122, 124, 218, 283, 331, 368, 370, 374, 377, 381, 434).

Line 29: new suggestion„Localization of all three Tffs was determined by ICH in differently aged human SM and healthy donors“ line 31 – 37 RESULTS  : please reformulate results to be more connected and fluent. For ex. sentence suggestion : Tff3 was expressed  in SM of  healthy donors and  disease affected OA and RA patients on mRNA and protein level.  Tff1 and Tff2 were not detected in SM of any monitored group. SF of all patients groups showed presence of all three TFF proteins  at different extent. 

Answer: The sentence has been changed accordingly.The abstract has dramatically reduced in number of words.

TFF1 had no significant difference among groups while…“ please express changes regarding disease conditionsfor ex: OA patient shad in comparison to healthy condition significantly reduced level of Tff3 and comparable Tff2 level. RA patients had increased level of Tff2   and less Tff3 .. etc.

Answer: The sentence has been changed accordingly.The abstract has dramatically reduced in number of words.

Please check the size of the Abstract and reduce it : max . is 200 words ?!

Answer: The abstract has dramatically reduced in number of words.

Line 64 remove text in brackets (also ITF) Line 90-93: please rewrite the sentence it is too big and not easy to follow.

Answer:We thank the reviewer for this hint. The sentence has been changed accordingly

Line 96 : reduce the title . Semi quantitative RT-PCR Line 100. ERROR ??? reference ???

Answer: The sentence has been changed accordingly.

Results should not be separated according used methods. Combine 2.1.1. and 2.1.2 .titles in one : suggestion: Presence and localization of TFFs in synovial membrane Combine Title 2.1.3 and 2.1.4. and since it is already combined in figure 2. And rename in : “Quantification of Tff3 in synovial membrane” or similar Lie 105 : use term TFFs for all three peptides or write them as TFF1 ,TFF2,TFF3 Line 111…

Answer: The sentence has been changed accordingly.

TFF1 and TFF2 were not present …instead “reveal negative results” Lines 115 ,125,147 ??? ERROR reference? Lines 122-148- formulate as one chapter according Figure 2. Line 134 new formulation of sentence C) TFF3 protein level in…… detected by ELISA Line 141 _: The protein level of TFF3 as only TFF detected in SM was determined by ELISA….rewrite further text in spirit of English grammar.

Answer: The sentence has been changed accordingly.

Data under Figure 2C would be better to present in form of table with range and mean values since diagram with Mean values is not informative enough .

Answer: We thank the reviewer for this comment. We have create table for the publication.

Line 149 : make shorter titel like : : Tffs in synovila fluid orPresence of Tffs in synovila fluid -

Answer: We can't follow the reviewer here. Because the title described in 1 sentence the result of the methods.

Lines 151-183: please do not show results separated by method: combine methods in one chapter as previously suggested and shown in Fig.3 Please present ELISA results in table as previously suggested for Fig.2.

Answer: We thank the reviewer for this comment. We have create table for the publication.

Line 152-160. Please address issue of protein size more pronounced .. 1. Describing specificity of Abs in stomach (Tff1 ? is missing in stomach ?, offer an explanation) and than present  the difference of protein size in CF. 

Answer: The sentence has been changed accordingly (line 139-149)

Discussion refers to this point (lines 248-254) , regarding modifications of proteins in SF but stress the issue of  Ab specificity in results. In Fig.3. A. Describe that  the size is  different than expected.

Answer: The sentence has been changed accordingly. See point 14 of the answer to Reviewer #3.

Include : in 187 line add text like: Detected difference in TFfs protein size imply additional proteinmodifications or similar so that it is immediately clear why the sizes do not match to expected ones.

Answer:The sentence has been changed accordingly (line 239-240).

Line 173 Please use active form :::” The results show” should be replaced with : “TFF2 protein conc. was 22,6 fold higher …. Figure 23. C _please check the statistical significance RA vs. healthy according diagrams that should be statistically relevant as well.

Answer: The sentence has been changed accordingly (line 162). The difference in protein concentration of TFF2 between healthy and RA in SF was checked again. Since the data ist not normally distributed we performed the non parametric Mann-Whitney-U-Test. The difference is not statistically significant. This can be explained by the fact that we see a really wide range in protein concentration of TFF2 in RA samples with some upward outliers. Looking at the median of the two groups (9,0 in healthy and 13,5 in RA) instead of the mean (13,7 in healthy and 203,5 in RA) we see, that this difference is clearly lower.

Lines 156,169,173181 etc. further on in text “Error.Reference ..” should be corrected References : ex. 4,5,8,10,39, 43 have issues regarding author names : please check it according journal propositions.

Answer: We thank the reviewer for this comment.

Round 2

Reviewer 1 Report

From a general point of view, the authors did not respond to any of the requests of this reviewer, except for the addition of 3 patients to the results shown in Figure 1, which was not even changed.
The authors justify their limitations with the difficulties of obtaining biological material from patients and although they welcome the suggestions they do not modify the results but present them in future studies.
This reviewer also clarifies to the authors the need to replicate the results on a larger sample, regardless of the difficulties and what is expected as a result, since the biological variability in patients and healthy donors is very wide.

However, this reviewer is aware of the difficulties and approve the manuscript beyond the limitations

Reviewer 2 Report

This reviewer does not find that the concerns raised by the reviewers have been addressed sufficiently